# Very Low-Density Lipoproteins of Metabolic Syndrome Modulates STIM1, Suppresses Store-Operated Calcium Entry, and Deranges Myofilament Proteins in Atrial Myocytes

**DOI:** 10.3390/jcm8060881

**Published:** 2019-06-20

**Authors:** Yi-Lin Shiou, Hsin-Ting Lin, Liang-Yin Ke, Bin-Nan Wu, Shyi-Jang Shin, Chu-Huang Chen, Wei-Chung Tsai, Chih-Sheng Chu, Hsiang-Chun Lee

**Affiliations:** 1Center for Lipid Biosciences, Kaohsiung Medical University Hospital, Kaohsiung 807, Taiwan; irpu10.yls@gmail.com (Y.-L.S.); hsintinglin2007@gmail.com (H.-T.L.); thwangg@gmail.com (L.-Y.K.); cchen@heart.thi.tmc.edu (C.-H.C.); jujuson993@gmail.com (C.-S.C.); 2Lipid Science and Aging Research Center, Kaohsiung Medical University, Kaohsiung 807, Taiwan; 3Division of Cardiology, Department of Internal Medicine, Kaohsiung Medical University Hospital, Kaohsiung Medical University, Kaohsiung 807, Taiwan; azygo91@gmail.com; 4Department of Pharmacology, College of Medicine, Kaohsiung Medical University, Kaohsiung 807, Taiwan; binnan@gap.kmu.edu.tw; 5Department of Internal Medicine, Faculty of Medicine, College of Medicine, Kaohsiung Medical University, Kaohsiung 807, Taiwan; sjshin@kmu.edu.tw; 6Institute/Center of Medical Science and Technology, National Sun Yat-sen University, Kaohsiung 807, Taiwan

**Keywords:** very low-density lipoprotein, metabolic syndrome, STIM1, SOCE, atrial myopathy, atrial fibrillation

## Abstract

Individuals with metabolic syndrome (MetS) are at high risk for atrial myopathy and atrial fibrillation. Very low-density lipoproteins (VLDLs) of MetS (MetS-VLDLs) are cytotoxic to atrial myocytes in vivo and in vitro. The calcineurin–nuclear factor of activated T-cells (NFAT) pathway, which is regulated by stromal interaction molecule 1 (STIM1)/ calcium release-activated calcium channel protein 1 (Orai1)–mediated store-operated Ca^2+^ entry (SOCE), is a pivotal mediator of adaptive cardiac hypertrophy. We hypothesized that MetS-VLDLs could affect SOCE and the calcineurin–NFAT pathway. Normal-VLDL and MetS-VLDL samples were isolated from the peripheral blood of healthy volunteers and individuals with MetS. VLDLs were applied to HL-1 atrial myocytes for 18 h and were also injected into wild-type C57BL/6 male mouse tails three times per week for six weeks. After the sarcoplasmic reticulum (SR) Ca^2+^ store was depleted, SOCE was triggered upon reperfusion with 1.8 mM of Ca^2+^. SOCE was attenuated by MetS-VLDLs, along with reduced transcriptional and membranous expression of STIM1 (*P* = 0.025), and enhanced modification of *O*-GlcNAcylation on STIM1 protein, while Orai1 was unaltered. The nuclear translocation and activity of calcineurin were both reduced (*P* < 0.05), along with the alteration of myofilament proteins in atrial tissues. These changes were absent in normal-VLDL-treated cells. Our results demonstrated that MetS-VLDLs suppressed SOCE by modulating STIM1 at the transcriptional, translational, and post-translational levels, resulting in the inhibition of the calcineurin–NFAT pathway, which resulted in the alteration of myofilament protein expression and sarcomere derangement in atrial tissues. These findings may help explain atrial myopathy in MetS. We suggest a therapeutic target on VLDLs to prevent atrial fibrillation, especially for individuals with MetS.

## 1. Introduction

Despite advances in current therapy, the prevention of atrial fibrillation remains a challenge. Changes in Ca^2+^ regulation and related processes are important mechanisms leading to the initiation and maintenance of atrial fibrillation [1]. The pathogenesis of early atrial myopathy remains unclear. Intracellular Ca^2+^ homeostasis is critical for normal cellular function, particularly for cardiomyocytes, and may be dysregulated in early atrial myopathy.

Store-operated Ca^2+^ entry (SOCE) is recognized to coexist with voltage-gated Ca^2+^ channels in cardiomyocytes and, though this is debated, is suggested to contribute to basal Ca^2+^ homeostasis [2]. A general function of SOCE is to replenish depleted sarcoplasmic reticulum (SR) Ca^2+^ stores [3]. It is commonly thought that beat-to-beat Ca^2+^ handling is mediated by excitation–contraction coupling. Nevertheless, upon facing different cardiac stressors, a sustained increase in intracellular Ca^2+^ takes place upon SOCE, which in turn activates the calcineurin–NFAT pathway, which is responsible for cardiac hypertrophy as an adaptive response [4]. Several studies have identified the essential role of SOCE in mediating NFAT nuclear translocation to develop cardiac hypertrophy induced either by IP3-generating agonists or by following pressure overload [2,5,6,7].

Stromal interaction molecule 1 (STIM1), a Ca^2+^-binding protein and a Ca^2+^ sensor in the SR, is capable of triggering SOCE by interacting with Ca^2+^ inward channels on the plasma membrane [8,9]. Stress-triggered STIM1 re-expression and consequent SOCE enhancement are critical upstream elements that facilitate an essential increase in cytosolic Ca^2+^ levels to control cardiac hypertrophy [4]. A hypothetical model of STIM1 activation is that STIM1 couples with calcium release-activated calcium channel protein 1, Orai1, the Ca^2+^ inward channel located in the plasma membrane, and forms an SR/plasma membrane junction as they mediate crucial Ca^2+^ communication between the SR lumen, the cytoplasm, and the extracellular space [9,10].

*O*-GlcNAcylation is a post-translational modification of protein in serine and threonine residues that occurs in the cytoplasm and nucleus. The addition and removal of O-GlcNAc is catalyzed by *O*-GlcNAc transferase and neutral β-*N*-acetylglucosaminidase, respectively [11]. There is compelling evidence for a pivotal role of *O*-GlcNAcylation in the regulation of Ca^2+^ signaling [12]. STIM1 is a target of *O*-GlcNAcylation, and increased *O*-GlcNAcylation modification of STIM1 has been shown to attenuate SOCE in cardiomyocytes [13,14]. *O*-GlcNAcylation is also important in the development of diabetic cardiomyopathy and hyperglycemia-induced arrhythmias [15,16]. Additionally, *O*-GlcNAcylation may integrate environmental and genetic information in disease pathologies such as metabolic syndrome (MetS) [17], with a potential mechanism linking the metabolic and calcium signaling pathways. Although it has been reported that *O*-GlcNAcylation is involved in cardiomyopathy correlated with insulin resistance [12,15,18], there have only been a few studies related to the role of *O*-GlcNAcylation of STIM1 and SOCE in atrial myopathy.

Our previous study demonstrated that very low-density lipoproteins (VLDLs) isolated from individuals with MetS (MetS-VLDLs) exert cardiac lipotoxicity and induce atrial remodeling and vulnerability to atrial fibrillation [19]. MetS-VLDLs are suggested to play a pivotal role in early atrial myopathy [19]. In addition, MetS-VLDLs induce *O*-GlcNAcylation on gap junction proteins and as a result causing intracardiac conduction delay [20]. We further hypothesized that STIM1-regulated SOCE and *O*-GlcNAcylation are involved in the pathophysiology of atrial myopathy for individuals with MetS. Therefore, the objective of this study was to assess how VLDLs affect STIM1 and SOCE in atrial myocytes. HL-1 cells derived from murine atrial myocytes have essential SOCE machinery with expression of STIM1 and channel Orai1 and are therefore useful for investigating SOCE in the atrium [21]. This study used HL-1 cells to assess the effects of MetS-VLDLs on STIM1 expression, the *O*-GlcNAcylation modification of STIM1, the regulation of SOCE and subsequent calcineurin–NFAT signaling, and myofilament protein expression.

## 2. Materials and Methods

### 2.1. VLDL Isolation

This study followed Helsinki Declaration principles, and the study was approved by the Kaohsiung Medical University Hospital Ethics Review Board. Blood was obtained from individuals who either met the criteria for MetS or were otherwise healthy and gave informed consent. Blood samples from five male MetS patients (55.0 ± 15.5 years old, all with type 2 diabetes mellitus, elevated triglyceride levels, and hypertension) and three healthy individuals (non-MetS subjects, 34.3 ± 9.3 years old) were used in this study. Normal-VLDLs (VLDL from non-MetS subjects) and MetS-VLDLs (*d* = 0.930–1.006 g/mL) were isolated by sequential ultracentrifugation as previously described [19,22,23]. Pooled VLDL samples were applied for subsequent experiments.

### 2.2. HL-1 Atrial Myocyte Culture and Incubation with Isolated VLDLs

Murine HL-1 atrial myocyte cells were maintained with fresh Claycomb medium in precoated culture flasks at 37 °C in a humidified atmosphere containing 5% CO_2_. Culture medium was supplemented with 87% Claycomb medium, 2 mM/L L-glutamine, 10% fetal bovine serum, 100 U/mL penicillin, 100 µg/mL streptomycin, and 0.1 mM/L norepinephrine, which is necessary for maintaining a differentiated cardiac phenotype in HL-1 culture [24]. The HL-1 cells were treated with 25 μg/mL specific VLDLs for 18 h. For experiments on testing the effects of the modulation of *O*-GlcNAcylation on STIM1 expression (see below), the inhibitor 6-Diazo-5-oxo-L-norleucine (5 μM/L) (DON, Sigma-Aldrich, St Louis, MO, USA) and the enhancer Thiamet G (10 µM/L) (Thm-G, Sigma-Aldrich, St Louis, MO, USA) were used.

### 2.3. Quantitative Real-Time Reverse Transcriptase PCR

Total RNA of HL-1 cells was prepared using TRI Reagent (Sigma-Aldrich, St Louis, MO, USA) and was then reverse-transcribed (Invitrogen, Carlsbad, CA, USA). Quantitative real-time RT-PCR was performed using an ABI 7500 real-time system (Applied Biosystems, Foster City, CA, USA) and TaqMan Universal Master Mix II (Applied Biosystems, Foster City, CA, USA) with TaqMan probe STIM1 (Mm01158413_m1; Thermo Fisher Scientific, Waltham, MA, USA) and Rn18S (Mm03928890_g1; Thermo Fisher Scientific, Waltham, MA, USA).

### 2.4. Isolation of Nuclear, Cytoplasmic, and Membrane Proteins from HL-1 Cells

The HL-1 cells were resuspended in 500 μL of ice-cold hypotonic buffer (10 mM HEPES, 0.5 mM dithiothreitol, 0.5 mM phenylmethylsulfonyl fluoride, and protease inhibitor (Sigma-Aldrich Corp., St. Louis, MO, USA), pH 7.9) and disrupted with 30 strokes of a tight-fitting Dounce homogenizer. The homogenate was centrifuged to remove the nuclei and mitochondria at 8000 rpm for 15 min. Afterward, the supernatant was centrifuged at 24,000 rpm for 30 min to remove cytoplasmic proteins. The membrane fraction was obtained as the pellet. The pellet underwent hypotonic buffer wash and was dissolved in 200 μL of hypotonic buffer. Membrane proteins were released through treatment with 0.25% Triton X-100 for 1 h.

### 2.5. Western Blot

Protein samples were separated using SDS-polyacrylamide gel and transferred to nitrocellulose membranes. Blots were incubated with 1:1000 anti-STIM1 (Cell Signaling, #4916, Danvers, MA, USA), 1:1000 anticalcineurin (Cell Signaling, #2641, Danvers, MA, USA), 1:500 anti-Orai1 (Santa Cruz, sc-68895, Dallas, TX, USA), 1:1000 anti-NFATc3 (Santa Cruz, sc-8321, Dallas, TX, USA), and 1:500 antiphosphorylated NFATc3 (Santa Cruz, sc-365785, Dallas, TX, USA). The membranes were then incubated with horseradish peroxidase-linked secondary antibody. The immunoblots were identified with SuperSignal West Picochemiluminescent substrate (Thermo Fisher Scientific, Waltham, MA, USA). Band intensity was calculated using ImageJ (National Institutes of Health), and intensity data from cytoplasmic, membranous, and nuclear proteins of interest were normalized to 1:1000 α-tubulin (Santa Cruz, sc-23948, Dallas, TX, USA), 1:1000 pan-cadherin (Sigma-Aldrich, C1821, St. Louis, MO, USA), and 1:500 lamin B (Santa Cruz, sc-6216, Dallas, TX, USA), respectively. Antidesmin (1:500, Upstate, Cat. #04-585, Lake Placid, NY, USA) was used for immunoblotting atrial tissue proteins.

### 2.6. Detection of O-GlcNAcylation STIM1 by Immunoprecipitation 

Protein extract was isolated from HL-1 cells using lysis buffer: 1 mg of protein was mixed with 1:1000 anti-STIM1 (Cell Signaling, #4916, Danvers, MA, USA) at 4 °C for 1 h and then incubated with Pierce™ protein A/G magnetic beads (Thermo Fisher Scientific, St. Peters, MO, USA) overnight at 4 °C. The immunoprecipitation matrix was washed twice with PBS containing 1% NP-40. The matrix-bound protein was eluted in sample buffer and then separated by 10% SDS-PAGE electrophoresis. The magnetic beads were saturated to capture all STIM1 proteins in the cell lysate. The immunoblot was done with 1:1000 anti-*O*-GlcNAc (MA1-076, Thermo Fisher Scientific, St Peters, MO, USA) and 1:1000 anti-STIM1 incubation overnight at 4 °C, and signals were detected by secondary antibodies with chemiluminescent visualization.

### 2.7. Measurement of Calcineurin Activity

The intracellular calcineurin activity of HL-1 cells was measured using an in vitro calcineurin phosphatase activity assay kit (Abcam, cat no. ab139461) according to the instruction protocols. The HL-1 cells were seeded to the 96-well plate, and the VLDLs were treated for 18 h. Whole-cell protein of HL-1 cells was determined and controlled for the activity assay. Calcineurin activity was calculated as the ratio of the phosphate release amount that was reflected as optical density, which was read on a microplate reader (Epoch Microplate Spectrophotometer, BioTek, Winooski, VT, USA) at wavelength of 620 nm.

### 2.8. Measurement of SR Calcium Load and SOCE

Prior to experiments (48–72 h), HL-1 cells (5.0 × 10^3^) were washed twice with PBS and plated on 35-mm optical dishes. For imaging, HL-1 cells were incubated with a Ca^2+^ indicator dye, fura-2-AM (Invitrogen, Waltham, MA), for 30 min. Furo-2-AM stock solution was made up in DMSO (1 mM/mL). For a working concentration of 2 μM/mL, 2 μL of stock was added to 1 mL of calcium-free Hank’s balanced salt solution (HBSS, Thermo Fisher Scientific, St Peters, MO, USA). Calcium imaging capture was performed on attached HL-1 cells on an Olympus Cell R system. To measure SOCE, dishes were perfused with 10 μM thapsigargin (TG) (Sigma-Aldrich, St Louis, MO, USA) in calcium-free HBSS (Thermo Fisher Scientific, St Peters, MO, USA) (0 mM Ca^2+^) to trigger SR Ca^2+^ release. At the peak of the TG response, dishes were perfused with 10 mM of caffeine (Caff) in HBSS (0 mM Ca^2+^) to test if the SR Ca^2+^ pool was depleted. The peak TG/Caff responses reflected the SR load. After the Ca^2+^ signals returned to a stable baseline, cells were reperfused with calcium content solution HBSS (1.8 mM Ca^2+^). SOCE was triggered upon reperfusion with 1.8 mM Ca^2+^ solution, and the peak response was measured. After SOCE declined, myocytes were tested for viability at the end with potassium chloride (KCl) (80 mM). Data were eliminated if the KCl response was less than a 100% increase above the baseline fluorescence. The data for each experiment were the average of 8 to 11 cells’ ratiometric changes in the same dish. Each group had five dishes. For the positive control groups, the STIM1 inhibitor SKF 96365 (Santa Cruz, Dallas, TX, USA) was applied (10 μM incubated for 48 h and 5 µM incubated for 72 h) before the SOCE measurement. The prolonged inhibition of STIM1 was chosen to be consistent with the experiments demonstrating the subsequent myofilament protein changes (shown below).

### 2.9. Tissue Protein Isolation from Mice Atrial Tissue

MetS-VLDLs or normal-VLDLs were administered by intravenous injection in 9-month-old wild-type C57BL/6 mice tail veins at a dose of 15 μg/g three times per week for 6 consecutive weeks, as in our previous study [19]. After anesthetization with 50–90 mg/kg of pentobarbital injected intraperitoneally, the mice chests were opened, their hearts were excised, and the atrial tissues were snap frozen in liquid nitrogen and stored in a freezer at −80 °C before use. Frozen tissue samples (*n* = 3 per group) were thawed for protein extraction. Mice atrial tissue samples were disrupted with 30 strokes of tissue grinder, and tissue protein extraction was isolated through incubation with 200 μL of lysis buffer on ice for 1 h. The homogenate was centrifuged at 12,000 rpm for 20 min. The protein concentration was calibrated with a Pierce™ BCA Protein Assay Kit (Thermo Fisher Scientific, St. Peters, MO, USA) at 37 °C for 30 min and was determined at 570 nm by an ELISA reader. All applicable institutional and governmental regulations concerning the ethical use of animals were conformed to, including the National Institutes of Health (NIH) guidelines being followed, and all animal procedures were approved by the Institutional Animal Care and Use Committee of Kaohsiung Medical University.

### 2.10. Myofilament and Contractile Protein Expression and Phosphorylation Analysis in VLDL-Treated HL-1 Cells and Atrial Tissues of VLDL-Injected Mice

Proteins were separated by electrophoresis on 1D-PAGE 15% polyacrylamide gels. Gels were loaded with an equal volume on each lane. Phosphorylated proteins were detected by Pro-Q^®^ Diamond stain following the manufacturer’s protocol (Invitrogen, Waltham, MA, USA). In short, the gels were fixed in 5% acetic acid and 50% methanol and incubated with Pro-Q^®^ Diamond stock solution in 25 mL of staining buffer for 1.5 h. The gel was scanned using a Typhoon 9400 (GE Healthcare, Chicago, IL, USA). Subsequently, total proteins were detected by SYPRO^®^ Ruby stain. The gels were fixed with 50% methanol and 7% acetic acid and stained with SYPRO^®^ Ruby stain overnight. Phospho-bands were identified according to the molecular weight, normalized to the total lane individually, and then averaged (*n* = 5 per group). For positive control groups, the STIM1 inhibitor SKF 96365 (Santa Cruz, Dallas, TX, USA) was applied (5 µM) for 72 h, and the calcineurin inhibitor FK506 (Sigma-Aldrich, St. Louis, MO, USA) was applied (30 μM) for 24 h. The same Pro-Q method was applied to the mice atrial tissue (*n* = 3 per group).

### 2.11. Transmission Electron Microscopy (TEM)

After rinsing with phosphate-buffered saline, small pieces of atrial tissue were immediately fixed in 2.5% glutaraldehyde in 0.1 M of Sørensen’s buffer at 4 °C. Following dehydration, samples were post-fixed in 1% osmium tetroxide (OsO4) and embedded in EPON 812 (Electron Microscopy Sciences, Hatfield, PA, USA). Ultrathin sections (60 nm) were stained with uranyl acetate and lead citrate. Images were captured on a Transmission Electron Microscope HT7700 (HITACHI, Tokyo, Japan).

### 2.12. Data Analysis and Statistics

Data are expressed as means ± SD unless indicated otherwise, and *n* indicates the number of cell samples. One-way ANOVA and Tukey’s multiple comparisons test were used to compare values between groups. For the experiments with small *n* numbers, nonparametric tests were also performed to confirm the statistical significance (Prism; GraphPad, San Diego, CA, USA). Statistical significance was considered to be a *P*-value ≤ 0.05.

## 3. Results

### 3.1. MetS-VLDLs, but Not Normal-VLDLs, Induced the Downregulation of STIM1 at the Transcriptional and Translational Levels in HL-1 Cells

First, the expression of two key components of SOCE in HL-1 cells and the effects of two different VLDLs on their expression were examined (Figure 1a–d). The effects of VLDLs on STIM1 expression were investigated by quantitative RT-PCR and western blot. Only MetS-VLDLs significantly reduced STIM1 in HL-1 cells, and the reduction was at both the transcriptional (0.81 ± 0.02-fold versus control, Figure 1a) and the translational levels (0.33 ± 0.03-fold versus control, Figure 1c). The protein expression of STIM1 was analyzed specifically in the membrane fraction. The degree of STIM1 reduction in membrane proteins and in RNA expression was different. Nevertheless, the expression of Orai1, which was only analyzed on the protein level, remained unchanged (normal-VLDLs 0.94 ± 0.17-fold versus control and MetS-VLDLs 0.95 ± 0.06-fold versus control, Figure 1d). The effect of MetS-VLDLs on *O*-GlcNAcylation in the whole cell was examined using whole-cell immunoblot (Figure 1g). The results showed that MetS-VLDLs enhanced *O*-GlcNAcylation in proteins extracted from compartments of the nucleus, cytosol, and membrane. To further determine whether enhanced *O*-GlcNAcylation alone could cause any change in STIM1 expression, the HL-1 cells were treated with a specific inhibitor (deoxynorleucine) and an enhancer (Thiamet G) of *O*-GlcNAcylation for 24 h (Figure 1g,h). The manipulation of *O*-GlcNAcylation did not change STIM1 expression. The results suggested that the MetS-VLDL-induced STIM1 reduction did not result from its effect on the modulation of *O*-GlcNAcylation.

### 3.2. MetS-VLDLs Enhanced the O-GlcNAcylation of STIM1 Proteins

Since it is known that *O*-GlcNAcylated STIM1 exerts less activation, we therefore assessed if MetS-VLDLs enhanced the *O*-GlcNAcylation of STIM1 proteins. The immunoprecipitation results (Figure 1e,f) showed enhanced *O*-GlcNAcylation in the MetS-VLDL group (1.70 ± 0.41-fold versus control), with no change in the normal-VLDL group (1.10 ± 0.44-fold versus control, *P* = 0.731). The *O*-GlcNAcylation blots revealed another strong band at 120 kDa that also changed in the MetS-VLDL group (Figure 1e). Since STIM1 and Orai1 can undergo unimolecular coupling [25], the 120-kDa band very likely represents the unimolecular coupling of STIM1 and Orai1.

### 3.3. MetS-VLDLs Suppressed SOCE in HL-1 Cells

To determine if SOCE can be suppressed by MetS-VLDLs along with the modulation of STIM1, the Ca^2+^ response was assessed in fluorescent Ca^2+^-labeled HL-1 cells (Figure 2). First, the SR store was assessed by the response of HL-1 upon administration of TG and caffeine to deplete the Ca^2+^ store in the SR. This response was demonstrated as an up-and-down wave using ratiometric tracing (Figure 2a–e). The peak of the TG/caffeine response was smaller in the MetS-VLDL group compared to the controls (Figure 2f). After the SR Ca^2+^ store was depleted, cells were perfused with 1.8 mM of Ca^2+^ solution. The SOCE was demonstrated as another up-and-down wave in ratiometric tracing. The peak was measured, and the average of 8–11 cells was obtained for each dish (experiment). The data analysis from five experiments for each group is shown in Figure 2f. The SOCE was significantly suppressed in the MetS-VLDL group (0.59 ± 0.03-fold versus control, *P* < 0.001). The SOCE remained unchanged in the normal-VLDL group under the same incubation conditions (0.86 ± 0.10-fold versus control, *P* = 0.093). A positive control experiment was performed through pre-incubation with the STIM1 inhibitor SKF 96365, which suppressed the SOCE. There was also a reduction in TG/caffeine-induced Ca^2+^ release, which presumably was a reduced SR Ca^2+^ store caused by the long-term (48 and 72 h) incubation with SKF 96365 (Figure 2d,e). In Figure 2f, SKF 96365 reduced the SR load without a complete depletion, suggesting a constitutive Ca^2+^ entry through a residual STIM1/Orai1 channel or others. The calcium release-activated channels (CRAC) channel inhibitor, which blocks STIM1–Orai1 coupling and induces more SOCE inhibition, was not applied in this study. The blockers for the voltage-dependent Ca^2+^ channel and Na/Ca exchanger have been shown to not affect the peak of SOCE in HL-1 cells [21]. These blockers were not applied in this study.

### 3.4. MetS-VLDLs Inhibited the Calcineurin–NFAT Pathway

To confirm if the calcineurin–NFAT pathway downstream of SOCE could be inhibited by MetS-VLDLs, western blots were used to assess the protein expression of cytosolic and nuclear calcineurin and NFAT (Figure 3a). In the MetS-VLDL group, nuclear expression of calcineurin was significantly reduced (0.48 ± 0.09-fold vs. control, Figure 3b), while cytosolic expression was unchanged (0.98 ± 0.53-fold vs. control, Figure 3c). Nuclear NFAT was significantly reduced by MetS-VLDLs (0.51 ± 0.05-fold vs. control, Figure 3d). Phosphorylated NFAT in cytosol was increased in the MetS-VLDL group (1.53 ± 0.5-fold vs. control, Figure 3e). Calcineurin activity was examined using a colorimetric assay (Figure 1f) and was significantly reduced in the MetS-VLDL group (0.71 ± 0.07-fold vs. control, *P* = 0.0001). All of the above changes were absent in the normal-VLDL group. These results revealed that MetS-VLDLs inhibited calcineurin-mediated NFAT dephosphorylation and NFAT nuclear importation and resulted in reduced calcineurin activity.

### 3.5. MetS-VLDLs Affected Myofilament Protein Expression and Caused Sarcomere Derangement

Given that the calcineurin–NFAT pathway mediates cardiac hypertrophy, we explored whether MetS-VLDLs could affect myofilament protein expression in HL-1 cells and in vivo as well. The VLDL injection did not cause a significant increase in body weight (control 32.3 ± 1.8, normal-VLDL 33.2 ± 1.9, MetS-VLDL 35.1 ± 3.9 g), but mice injected with VLDLs extracted from MetS patients had a significantly increased heart weight at the time of sacrifice (control 0.22 ± 0.04, normal-VLDL 0.19 ± 0.11, MetS-VLDL 0.30 ± 0.04 g, *P* < 0.01 for control vs. MetS-VLDL) (Appendix A). The observation of heart chamber enlargement (left atria and left ventricles) and a reduced left ventricular ejection fraction (Appendix A) was consistent with our previous study [18]. There was no significant change in circulatory triglyceride and total cholesterol. The data from biochemistry analysis of the animal blood samples collected at their sacrifice are shown in Appendix A.

Because the murine atrial tissue was too small to perform multiple protein western blotting, ProQ Diamond Stain in combination with SYPRO stain was used. This method enables simultaneous quantitative analysis of multiple contractile and myofilament phosphoproteins using small amounts of atrial tissue [26]. The expression of phosphorylated troponin I (TnI) and troponin T (TnT) was increased in the MetS-VLDL groups of HL-1 cells and atrial tissues (Figure 4b,d). The expression of phosphorylated myosin regulatory light chain 2 (MLC2) was reduced in MetS-VLDL groups (Figure 4b,d). The expression of desmin was increased in the atrial tissues (Figure 4d), and the result was consistent in immunoblotting using the specific antibody (Figure 4e,f). These changes were consistent with changes upon administration of the STIM1 inhibitor SKF 96365 (72 h incubation) or the calcineurin inhibitor FK506 (24 h incubation) (Figure 4a,b). The opposite change was shown for myosin-binding protein C (cMyBPC), which was significantly reduced in MetS-VLDL-injected mice (msVLDL in Figure 4d) atrial tissues but not in HL-1 cells after MetS-VLDL treatment. Of note, there was no change in myofilament protein expression with short durations of SKF96365 treatment (24 h). It is noteworthy that in addition to targeting STIM1, SKF96365 has been found to exhibit nonspecific inhibition in transient receptor potential-canonical (TRPC) channels, voltage-gated Ca^2+^ channels, and some potassium channels [27]. Therefore, the possibility of nonspecific inhibition from SKF96365 on mediating other channels to affect SOCE in the present study cannot be excluded.

To see if alteration of myofilament protein expression coexisted with sarcomere derangement, TEM was applied. The images showed disorganized Z lines of sarcomeres in atrial tissues of mice receiving a MetS-VLDL injection (Figure 4e). The results indicated that MetS-VLDLs altered atrial myofilament proteins and induced sarcomere derangement.

## 4. Discussions

The key findings of this study were that MetS-VLDLs reduced STIM1 on the transcriptional and translational levels and enhanced the *O*-GlcNAcylation modification on STIM1. VLDL-induced STIM1 modulation in turn suppressed SOCE and the downstream calcineurin–NFAT pathway, along with alterations of myofilament proteins and sarcomere derangement in mouse atrial tissues.

It has been suggested that STIM1-Orai1-dependent SOCE assists in maintaining intracellular Ca^2+^ in HL-1 cardiomyocytes [21]. Consistently, STIM1 and Orai1 expression was clearly demonstrated in our experiments through western blots, and SOCE could be assessed when cells were re-perfused with a physiological concentration of Ca^2+^ solution after SR Ca^2+^ depletion by TG and caffeine. Coinciding with this manuscript preparation, there have been more recent reports regarding the regulation of STIM1 expression [28,29,30]. The importance of SOCE and STIM1 signaling in heart physiology has also been more and more noticed [31]. Although the present study did not determine the specific mechanism by which the MetS-VLDLs induced Orai1-independent STIM1 suppression, the results contribute some understanding to how SOCE could act as a mediator in linking the pathogenesis from VLDLs to atrial myopathy.

A link between the regulation of cellular *O*-GlcNAcylation and Ca^2+^ signaling has been proven [13]. In Zhu-Mauldin’s study, increased *O*-GlcNAcylation of STIM1 prevented STIM1 puncta formation and blunted STIM1-mediated SOCE. The observation of enhanced *O*-GlcNAcylation of STIM1 was coincidental with the MetS-VLDL-suppressed SOCE, suggesting that *O*-GlcNAcylation of STIM1 may affect SOCE. On the other hand, *O*-GlcNAcylation per se did not affect the protein expression of STIM1 in the present study. *O*-GlcNAcylation synthesis has been thoroughly proven to impair SOCE-mediated transcription in hyperglycemia and diabetes [15,32,33,34]. For the first time, this study reports MetS-VLDLs as mediators in enhancing the *O*-GlcNAcylation of STIM1. Additional studies are needed to better understand how MetS-VLDLs enhance *O*-GlcNAcylation in atrial cardiomyocytes. This understanding may reveal a novel mechanism linking the metabolic and calcium signaling pathways in cardiac lipotoxicity.

In addition to mediating cardiac hypertrophy, short-term adaptation of energy metabolism enzymes to mechanical loads was shown to critically depend on the calcineurin pathway [35]. The calcineurin–NFAT pathway is suggested to be critical in both pathological hypertrophy and cardiac adaptation to biomechanical stress [35]. In patients and animal models with atrial fibrillation, calcineurin activity and expression are increased [36]. The upregulation of calcineurin is suggested to occur in response to atrial fibrillation-related tachycardia [37], which is different from our study, which identified calcineurin changes in the lipotoxicity of VLDLs at a very early stage of atrial cardiomyopathy. For a reduction of overall calcineurin protein, more studies are needed to elucidate other regulatory pathways. NFAT can also be phosphorylated by a large family of kinases, such as JNK, p38, GSK3β, casein kinase I and II, protein kinase, and possibly also ERK [38]. We suggest that the reduction of nucleus NFAT by MetS-VLDLs resulted from suppressed SOCE-calcineurin-mediated dephosphorylation.

Changes in sarcomere proteins in myofilaments have been shown to be associated with different pathophysiological conditions, such as aging hearts, cardiac hypertrophy, heart failure, and cardiomyopathies [39]. Our assessment confirmed that MetS-VLDLs could influence myofilament protein expression (Figure 4a–d). These changes were consistent with changes upon administration of the STIM1 inhibitor SKF 96365, which also suppressed SOCE in the same incubation conditions (Figure 4a,b and Figure 2d,e). Sarcomere organization and integrity are controlled by complex and dynamic mechanisms [40]. Moreover, the remodeling of myofilament phosphorylation in response to atrial fibrillation and atrial dilatation is complicated [37]. Further studies are needed to elucidate changes in myofilament phosphorylation at a different stage of VLDL-induced atrial cardiomyopathy.

The MetS-VLDLs were isolated from blood samples of patients who had been receiving regular and appropriate medical treatment. The management of treatment was basically in accordance with clinical guidelines, with a goal set of HbA1c in the range of 6.5% to 7% (estimated average glucose level 140 to 154 mg/dL), LDL cholesterol <100 mg/dL, and blood pressure ≤130/80 mmHg. A lipid-lowering agent was not prescribed for elevated triglyceride unless the level was over 500 mg/dL. Although it was not determined how the presence of diabetes mellitus contributed to the distinct effects from the MetS-VLDLs, we believe the results from MetS-VLDLs in the present study were reflective of MetS. This can probably be proven by comparing samples between diabetes alone (noncomplicated type 1 diabetes mellitus) and diabetes with combined MetS.

Some limitations need to be addressed. First, the alterations in the SOCE signaling pathway were not the same in the HL-1 cells and in the atrial tissue with different durations of VLDL application (72 h for the in vitro and 6 weeks for the in vivo). Second, immunoprecipitation was not feasible with the limited protein sample amount (to examine *O*-GlcNAcylation in murine atrial tissue). Third, we did not examine other calcium regulation proteins, such as Cav 1.2, that can possibly be modulated upon chronic exposure to VLDLs or the STIM1 inhibitor SKF. It remains undetermined how other calcium-related channels or proteins intervene in the effects of MetS-VLDLs on the SOCE signaling pathway. Another limitation is the absence of data on other mediators of SOCE, such as STIM2, Orai2, and Orai3; data on the coupling of STIM1 with Orai1/3; and data on other voltage-dependent Ca^2+^ channels and TRPC channels, which have recently been reported with potential arrhythmogenic roles in mouse ventricular myocytes [41]. Lastly, the concentrations of total cholesterol, low-density-lipoprotein-cholesterol, and VLDL-cholesterol in mice are lower than in humans. Therefore, the biochemistry data (Appendix A) cannot be extrapolated to humans.

### Clinical Implications

Considering the vulnerability of MetS individuals to develope atrial fibrillation, this study sheds some light on the pathogenesis of VLDL-mediated atrial remodeling in MetS (Figure 5). MetS-VLDLs induced alterations in myofilament protein expression along with the suppression of SOCE. MetS-VLDLs have been shown to induce excessive lipid accumulation, atrial remodeling, and delayed intra-atrial conduction. Moreover, mice injected with MetS-VLDLs developed atrial fibrillation [19,20]. In a human study of patients with valve heart diseases and diabetes, calcineurin–NFATc3 signaling was shown to correlate with the presence of atrial fibrillation [36]. It is possible that STIM1–SOCE regulation changes during the progression of early atrial myopathy, even turning into a transition after the stage of persistent atrial fibrillation. Before the occurrence of atrial fibrillation, we suggest that VLDL triggers and enhances the progression of atrial remodeling (Figure 5). Human studies are mandatory to elucidate whether currently available lipid-lowering agents, such as fibrates, or any other compounds can reduce the cardiac lipotoxicity from VLDL-mediated lipid accumulation to improve atrial cardiomyopathy and to prevent atrial fibrillation in MetS.

## 5. Conclusions

MetS-VLDLs reduced STIM1 expression at the transcriptional and translational levels. MetS-VLDLs also modified STIM1 via enhanced *O*-GlcNAcylation. This STIM1 modulation suppressed SOCE and inhibited the Ca^2+^–calcineurin–NFAT pathway, resulting in the alteration of sarcomere protein expression in atrial myocytes. These findings may partially explain the pathogenesis of atrial myopathy in MetS. For controlling atrial myopathy in its progression to prevent atrial fibrillation, we suggest a therapeutic target on VLDLs, especially for individuals with MetS.

## Figures and Tables

**Figure 1 jcm-08-00881-f001:**
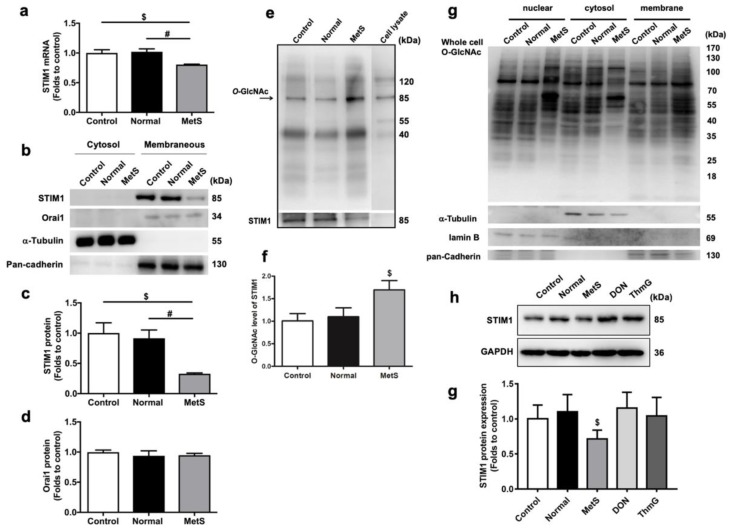
Effects of metabolic syndrome (MetS)-very low-density lipoproteins (VLDLs) on the expression of stromal interaction molecule 1 (STIM1) and calcium release-activated calcium channel protein 1 (Orai1) and the *O*-GlcNAcylation of STIM1. (**a**) Quantitative RT-PCR of STIM1 (*n* = 4 for each group). Reduced STIM1 mRNA in the MetS-VLDL-treated group (MetS) (^$^
*P* = 0.012, ^#^
*P* = 0.005). (**b**) Representative bands of western blots for STIM1 and Orai1 channel proteins. (**c**) Reduced STIM1 membrane protein expression in the MetS-VLDL group (*n* = 4 for each group; ^$^
*P* = 0.025, ^#^
*P* = 0.021). (**d**) Orai1 channel membrane protein expression among groups (*n* = 4 for each group; *P* = 0.5223). (**e**,**f**) Representative immunoblots and densitometry analysis (*n* = 4 for each group). Although STIM1 protein expression was reduced, the *O*-GlcNAcylation (85 kDa, indicated by the arrow) was larger in the MetS group. ^$^
*P* = 0.038 for the MetS group versus the control. All of the changes in the MetS group were absent in the normal-VLDL group (Normal). (**g**) Whole-cell *O*-GlcNAcylation immunoblotting showed enhanced *O*-GlcNAcylation of the nuclear, cytosol, and membranous protein fractions in the MetS groups. (**g**,**h**) The inhibition (with 5 μM deoxynorleucine (DON)) and enhancement (with 10 nM Thiamet G (ThmG)) of *O*-GlcNAcylation did not affect the expression of STIM1.

**Figure 2 jcm-08-00881-f002:**
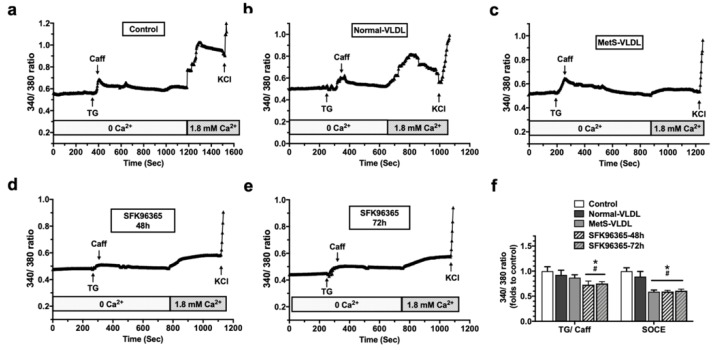
Suppressed store-operated Ca^2+^ entry (SOCE) following sarcoplasmic reticulum (SR) Ca^2+^ depletion in MetS-VLDL-treated HL-1 cardiomyocytes. (**a**–**c**) Representative ratiometric tracings in fluorescence measurements in control, normal-VLDL, and MetS-VLDL-treated HL-1 cells during SOCE testing. (**d**–**e**) Representative ratiometric tracings from STIM-1 inhibited HL-1 cells. SKF 96365, a STIM1 inhibitor. (**f**) Analysis of data from the ratiometric fluorescence 340:380 ratio for the peak response to thapsigargin/caffeine (TG/Caff) (*n* = 5 experiments; ^$^
*P* = 0.024, ^#^
*P* < 0.001, * *P* < 0.01, all vs. control) and the peak of SOCE (*n* = 5 experiments; all ^#^
*P* < 0.001 vs. control; all * *P* < 0.001 vs. normal-VLDLs).

**Figure 3 jcm-08-00881-f003:**
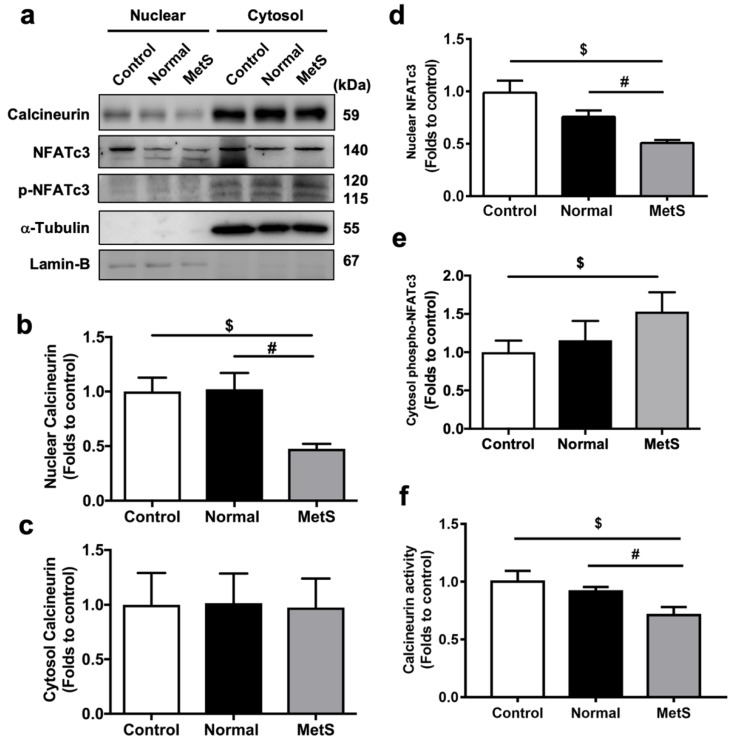
MetS-VLDLs suppressed calcineurin–nuclear factor of activated T-cells (NFAT) signaling pathways. (**a**) Representative bands of western blots (*n* = 4 for each group) for nuclear and cytosolic fractions of calcineurin, NFAT, and phosphorylated NFAT in proteins. (**b**) Reduced nuclear calcineurin in the MetS-VLDL group (MetS) (*n* = 4; ^$^
*P* = 0.037 vs. control, ^#^
*P* = 0.04 vs. normal-VLDL group (Normal)). (**c**) Unchanged cytosolic expression of calcineurin protein (*n* = 4, *P* = 0.9377). (**d**) Reduced nuclear NFAT in the MetS group (*n* = 4; ^$^
*P* = 0.007, ^#^
*P* = 0.009). (**e**) Increased phosphorylated NFAT in the cytoplasm of the MetS group (*n* = 4, ^$^
*P* = 0.04). (**f**) Reduced calcineurin activity in the MetS group (*n* = 6; ^$^
*P* = 0.0001, ^#^
*P* = 0.0001).

**Figure 4 jcm-08-00881-f004:**
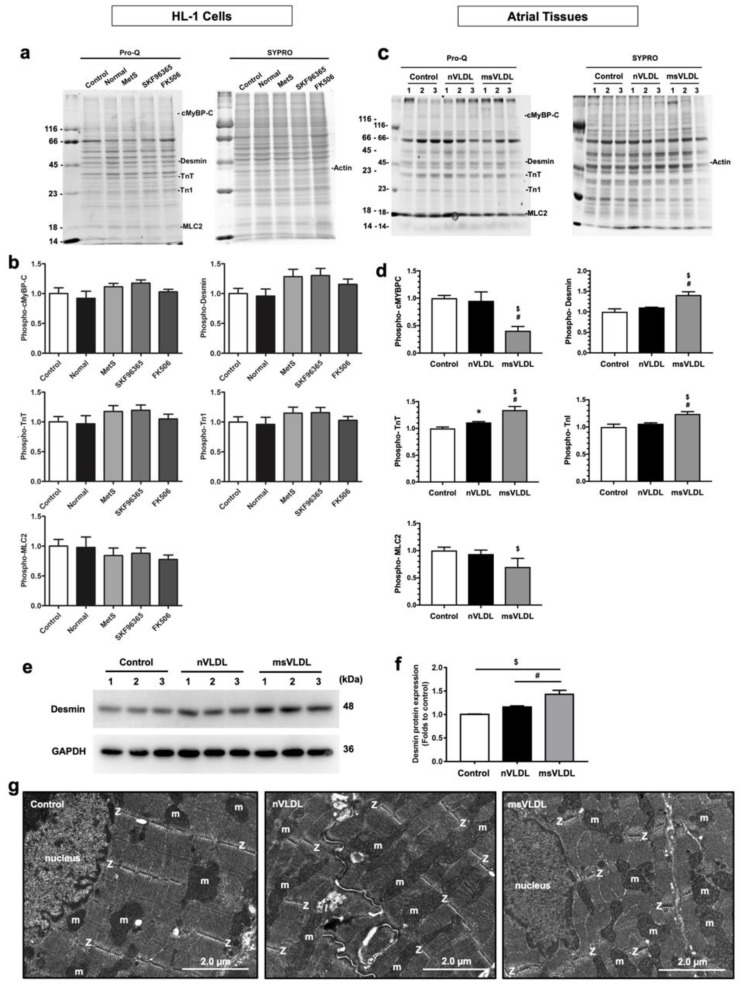
MetS-VLDLs altered myofilament protein expression and induced sarcomere derangement. (**a**) Phosphorylated cardiac myosin-binding protein C (cMyBPC), cardiac troponin I (TnI) and T (TnT), and myosin light chain 2 (MLC2) on 1D-gradient gels stained with ProQ Diamond reagent (left) and total protein expression shown on the gel subsequently stained with SYPRO Ruby (right). SKF 96365, a STIM1-inhibtor; FK-506, a calcineurin-inhibitor; *n* = 4 for each group. (**b**). Densitometry analyses from gels with ProQ and SYPRO staining. (**c**) ProQ Diamond staining gels of mouse atrial proteins (*n* = 3 for each group: Control; Normal-VLDL-injected mice, nVLDL; and MetS-VLDL-injected mice, msVLDL). (**d**) Densitometry analyses for phosphorylated cMyBPC, desmin, TnT, TnI, and MLC2. ^$^
*P* < 0.05 msVLDL versus control; ^#^
*P* < 0.05 msVLDL versus nVLDL. (**e**,**f**) Western blot and densitometry analysis for desmin in atrial tissues. (**g**) Representative transmission electron microscopy (TEM) pictures (at 5000× magnification) showing disorganized Z lines (Z, highlighted with dashed lines) in the atrial tissue of mice receiving a MetS-VLDL injection (msVLDL) compared to normal Z lines in the controls and in mice receiving normal-VLDLs (nVLDL) (*n* = 3 for each group). Atrial sarcomeres are aligned with mitochondria (m).

**Figure 5 jcm-08-00881-f005:**
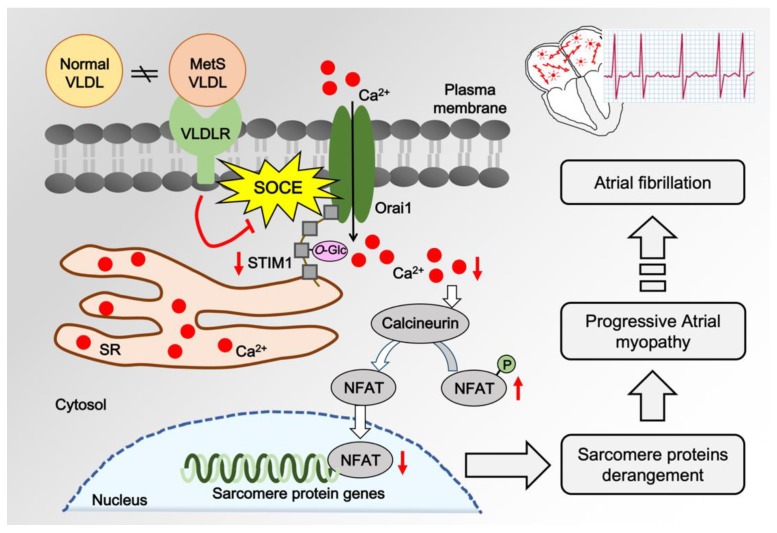
Lipotoxicity of VLDLs on mediating maladaptation of calcium regulation to derangement of sarcomere proteins in atrial myopathy. In metabolic syndrome (MetS), VLDLs undergo biochemical property changes and become different from VLDLs of normal conditions [23]. MetS-VLDLs reduce STIM1 expression and enhance *O*-GlcNAcylation on STIM1 protein. These changes in concert suppress SOCE and the downstream calcineurin–NFAT pathway, resulting in alteration of myofilament protein expression, disruption of sarcomere organization, and atrial myopathy [35]. The progression of atrial myopathy ultimately leads to atrial fibrillation.

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
