# Peer review of "Very Low-Density Lipoproteins of Metabolic Syndrome Modulates STIM1, Suppresses Store-Operated Calcium Entry, and Deranges Myofilament Proteins in Atrial Myocytes"

_jcm, 2019, doi:10.3390/jcm8060881_

Reviewer 1 Report

The paper is very well written, and the topic is of great interests. Cellular pathways mediated by TRPC channels are earning a growing interest in the cardiological community for several reasons. The potential involvement of SOCE suppression in the early stages of atrial cardiomyopathy may contribute to consistently increase our understanding of the mutual interplay between metabolic syndromes and atrial arrhythmias susceptibility. 

The reviewer has only minor comments:

Comment 1: Page 309: The authors report about a significant increase in body weight in mice following MetS-VLDL injection, and subsequently describe the presence of both left atrial and left ventricular enlargement. Supplementary table number 1 could be enriched with additional information about left atrial and left ventricular wall thickness in the 3 group and, if available, blood pressure measurements in the three groups (controls, nVLDL and Met-VLDL). These additional data appear of particular interest due to the known and very well described by the authors involvement of SOCE in adaptive cardiac hypertrophy.

Comment 2: Page 2, line 326, typo: MetVLDL, not msVLDL

Comment 3: The Potential Arrhythmogenic Role of TRPC Channels and SOCE Mechanism has recently been reported at the ventricular level as well (Wen H,Zhao Z,Fefelova N,Xie LH.Potential Arrhythmogenic Role of TRPC Channels and Store-Operated Calcium Entry Mechanism in Mouse Ventricular Myocytes. Front Physiol. 2018 Dec 13; 9:1785. doi: 10.3389/fphys.2018.01785. eCollection 2018). The authors should very briefly describe these findings in the discussion, to further underline the biological complexity of  TRPC mediated pathway at the cardiac level.

Author Response

Response to Reviewer 1 Comments

Comments and Suggestions for Authors

The paper is very well written, and the topic is of great interests. Cellular pathways mediated by TRPC channels are earning a growing interest in the cardiological community for several reasons. The potential involvement of SOCE suppression in the early stages of atrial cardiomyopathy may contribute to consistently increase our understanding of the mutual interplay between metabolic syndromes and atrial arrhythmias susceptibility. 

Response: We would like to thank the Reviewer for pointing out the significance of this work and for the expert and very helpful comments. In the following we respond to the comments.

The reviewer has only minor comments:

Comment 1: Page 309: The authors report about a significant increase in body weight in mice following MetS-VLDL injection, and subsequently describe the presence of both left atrial and left ventricular enlargement. Supplementary table number 1 could be enriched with additional information about left atrial and left ventricular wall thickness in the 3 group and, if available, blood pressure measurements in the three groups (controls, nVLDL and Met-VLDL). These additional data appear of particular interest due to the known and very well described by the authors involvement of SOCE in adaptive cardiac hypertrophy.

Response 1: Thank you for this comment. The echocardiographic measurements for left atrial and ventricular size, wall thickness, and ejection fraction of left ventricle had been added to the Supplementary Table 1. We did not measure blood pressure in the three groups. The additional statement had been added to Results (line 320-321). 

Comment 2: Page 2, line 326, typo: MetVLDL, not msVLDL

Response 2: Thank you for this comment. “msVLDL” had been replaced to “MetS-VLDL”. For readers’ better understanding, the Normal-VLDL and MetS-VLDL injected mice are referred to nVLDL and msVLDL, respectively in Figure 4. 

Comment 3: The Potential Arrhythmogenic Role of TRPC Channels and SOCE Mechanism has recently been reported at the ventricular level as well (Wen H,Zhao Z,Fefelova N,Xie LH.Potential Arrhythmogenic Role of TRPC Channels and Store-Operated Calcium Entry Mechanism in Mouse Ventricular Myocytes. Front Physiol. 2018 Dec 13; 9:1785. doi: 10.3389/fphys.2018.01785. eCollection 2018). The authors should very briefly describe these findings in the discussion, to further underline the biological complexity of TRPC mediated pathway at the cardiac level.

Response 3: Thank you for this comment. We had read Wen H et al. paper and it had been cited as reference #41 with related statements added to the Discussion (Line 426~ 427) as following: “…and TRPC channels, which had recently been reported with potential arrhythmogenic role in mouse ventricular myocytes [41].”

Reviewer 2 Report

This manuscript by Lee et al. focused on the pathogenesis of atrial fibrillation (AF) in patients with metabolic syndrome (MetS). Authors demonstrated that MetS-VLDL suppresses SOCE by modulating STIM1, resulting in inhibition of the calcineurin-NFAT pathway that results in alteration of myofilament protein expression and sarcomere derangement in atrial tissues. As authors mentioned, these findings can be possible etiology on the high frequency of atrial myopathy in MetS. Although this manuscript seems written very well, authors may want to resolve minor issues as follows.

 Minor comments;

1) In conclusion, for controlling atrial myopathy from progression to prevent AF, authors suggested that therapeutic target on VLDL, especially for individuals with MetS. This concept seems agreeable. However, authors did not describe how physicians should manage patients with MetS, or what drug is better for this purpose. For instance, should we use fibrate to prevent AF in patients with MetS?

2) In Supplementary Table 1, “nVLDL” and “msVLDL” should be described as “normal-VLDL” and “MetS-VLDL”, as described in the main text.

3) According to the data in Supplementary Table 1, concentrations of total cholesterol, LDL-C, and VLDL-C were different from those in human (lower than those in human). Therefore, these results may not always extrapolate in human. Authors may want to comment it.

Author Response

Response to Reviewer 2 Comments 

Comments and Suggestions for Authors

This manuscript by Lee et al. focused on the pathogenesis of atrial fibrillation (AF) in patients with metabolic syndrome (MetS). Authors demonstrated that MetS-VLDL suppresses SOCE by modulating STIM1, resulting in inhibition of the calcineurin-NFAT pathway that results in alteration of myofilament protein expression and sarcomere derangement in atrial tissues. As authors mentioned, these findings can be possible etiology on the high frequency of atrial myopathy in MetS. Although this manuscript seems written very well, authors may want to resolve minor issues as follows.

Response: We would like to thank the reviewer for pointing out the main findings of this work and for the expert and very helpful comments. In the following we respond to the comments.

 Minor comments;

Point 1)In conclusion, for controlling atrial myopathy from progression to prevent AF, authors suggested that therapeutic target on VLDL, especially for individuals with MetS. This concept seems agreeable. However, authors did not describe how physicians should manage patients with MetS, or what drug is better for this purpose. For instance, should we use fibrate to prevent AF in patients with MetS? 

 Response 1: Thank you for this comment. We added the following statement to the Discussion (line 440-443): “Human studies are mandatory to elucidate whether currently available lipid-lowering agents, such as fibrates or any other compounds can reduce the cardiac lipotoxicity from VLDL-mediated lipid accumulation to improve atrial cardiomyopathy and to prevent atrial fibrillation in MetS.”

 Point 2) In Supplementary Table 1, “nVLDL” and “msVLDL” should be described as “normal-VLDL” and “MetS-VLDL”, as described in the main text.

 Response 2: Thank you for this comment. “nVLDL” and “msVLDL” had been replaced to “Normal-VLDL” and “MetS-VLDL”, respectively. For readers’ better understanding, the Normal-VLDL and MetS-VLDL injected mice are referred to nVLDL and msVLDL, respectively in Figure 4.

 Point 3)According to the data in Supplementary Table 1, concentrations of total cholesterol, LDL-C, and VLDL-C were different from those in human (lower than those in human). Therefore, these results may not always extrapolate in human. Authors may want to comment it.

 Response 3: Thank you for this comment. We added the following statement to the Discussion (line 427-429). “Lastly, concentrations of total cholesterol, LDL-C, and VLDL-C of mice were lower than those in human. Therefore, the biochemistry data (Supplementary Table 1) cannot be extrapolated in human”.